# AI-based analysis of oral lesions using novel deep convolutional neural networks for early detection of oral cancer

**Kritsasith Warin**[1]*, **Wasit Limprasert**[2], **Siriwan Suebnukarn**[1], **Suthin Jinapornthham**[3], **Patcharapon Jantana**[4], **Sothana Vicharueang**[4]

**1** Faculty of Dentistry, Thammasat University, Khlong Luang, Pathum Thani, Thailand, **2** College of Interdisciplinary Studies, Thammasat University, Khlong Luang, Pathum Thani, Thailand, **3** Faculty of Dentistry, Khon Kaen University, Nai Muang, Khon Kaen, Thailand, **4** StoreMesh, Thailand Science Park, Khlong Luang, Pathum Thani, Thailand

* warin@tu.ac.th

**Data Availability Statement:** The data for this study cannot be shared publicly because of ethical and institutional regulations imposed by the

## Abstract

Artificial intelligence (AI) applications in oncology have been developed rapidly with reported successes in recent years. This work aims to evaluate the performance of deep convolutional neural network (CNN) algorithms for the classification and detection of oral potentially malignant disorders (OPMDs) and oral squamous cell carcinoma (OSCC) in oral photographic images. A dataset comprising 980 oral photographic images was divided into 365 images of OSCC, 315 images of OPMDs and 300 images of non-pathological images. Multiclass image classification models were created by using DenseNet-169, ResNet-101, SqueezeNet and Swin-S. Multiclass object detection models were fabricated by using faster R-CNN, YOLOv5, RetinaNet and CenterNet2. The AUC of multiclass image classification of the best CNN models, DenseNet-196, was 1.00 and 0.98 on OSCC and OPMDs, respectively. The AUC of the best multiclass CNN-base object detection models, Faster R-CNN, was 0.88 and 0.64 on OSCC and OPMDs, respectively. In comparison, DenseNet-196 yielded the best multiclass image classification performance with AUC of 1.00 and 0.98 on OSCC and OPMD, respectively. These values were inline with the performance of experts and superior to those of general practitioners (GPs). In conclusion, CNN-based models have potential for the identification of OSCC and OPMDs in oral photographic images and are expected to be a diagnostic tool to assist GPs for the early detection of oral cancer.

## Introduction

The power and potential of artificial intelligence (AI) innovations in healthcare are increasingly proven by the desire to improve the quality of clinical care. Novel AI technologies can help clinicians reduce human errors and increase the accurate decision-making with superior outcomes compared to traditional methods [1]. AI applications in head and neck cancer diagnosis have been developed rapidly with reported successes in the initial interpretation of medical images [2]. Among the technological advancements in AI, deep convolutional neural networks (CNN) are the algorithms based on neural networks that mimic the mechanism of

Human Research Ethics Committee of Thammasat University. Applications to access the data used in this research must be reviewed and approved by the Human Research Ethics Committee of Thammasat University prior to being shared to researchers. Data requests can be made by contacting ecsctu3@tu.ac.th.

**Funding:** This study was supported by the Thammasat University Research Grant (TUFT24/2564) and the Health Systems Research Institute, Thailand (Grant No. 65-025). The funders had no role in study design, data collection and analysis, decision to publish, or preparation of the manuscript.

**Competing interests:** The authors have declared that no competing interests exist.

human neurons. CNNs are currently being developed as tools to assist clinicians in solving various problems and to increase the accuracy of disease detection in radiographic images and clinical images [3]. The CNN-based algorithms, such as faster R-CNN, ResNet, and DenseNet, have been used to detect and classify lesions in chest x-rays [4] and lesions from clinical images of the skin, cervix, esophagus and larynx, with expert level results [5–8]. The advent of AI technology does not mean the ultimate replacement of clinicians. Instead, it will help clinicians, especially general practitioners (GPs), evaluate and diagnose patients more accurately.

According to the global cancer situation, cancer of the oral cavity, like other life-threatening diseases, is a highly relevant global public health problem. Although oral cancers are the 18th most common cancer worldwide, they are a fatal disease which caused over 170,000 deaths in the year 2020 [9]. Oral squamous cell carcinoma (OSCC) is one frequent malignancy in the oral cavity which accounts for about 90% of all oral cancers [10]. Two-thirds of oral cancers have been found in developing and low to middle income countries, especially in Southeast Asia and South Asia [9]. Most cases of OSCC are transformed from oral potentially malignant disorders (OPMDs) of the oral cavity such as erythroplakia, leukoplakia, erythroleukoplakia, oral lichen planus, etc., which have approximately a 1% potential to transform into a malignancy lesion [11]. OPMDs and early stages of OSCC are often asymptomatic and may appear as harmless lesions so they may be easily misrecognized, especially by general practitioners (GPs) [12], which leads to delayed diagnosis. Treatment of oral cancer depends on the cancer staging. The advance stages of oral cancer often involves more invasive treatment which increases morbidity, cost of treatment and significantly impacts the individual's quality of life [13, 14]. The prognosis of oral cancer worsens in the advanced stages of cancer. The 5-year survival rate of early stage oral cancer is approximately 69.3% but will decrease to 31.2% in the advanced stage [15, 16]. This number has not significantly improved in the past few decades regardless of various treatments [10]. In addition, the cost of treating oral cancer is extremely high, especially in the late stage, which is higher than that of OPMDs and in the early stage approximately 7.25 and 2.75 times, respectively [17]. Therefore, the early detection could reduce the economic burden of oral cancer.

Early detection oral cancer, is therefore very important as it not only increases the survival rate but also improves the quality of life of patients. The aim of this study is to evaluate the performance of CNN-based algorithms for the classification and detection of OPMD and OSCC in oral photographic images, and compare the automatic classification performance of these algorithms to experts (board-certified oral and maxillofacial surgeons) and GPs. These automatic models, combined with clinical data, are expected to provide a new diagnostic tool for GPs to improve the accuracy of early detection of cancerous lesions and to support expert-level decision making in the oral cancer screening program.

## Materials and methods

### Data description

This study was approved by the Human Research Ethics Committee of Thammasat University (COE 020/2563) and was performed in accordance with the tenets of the Declaration of Helsinki. Informed consent was waived because of the retrospective nature of the fully anonymized images. All clinical oral photographs analyzed in this study were collected retrospectively from the Oral and Maxillofacial Surgery Center of Thammasat University and Khon Kaen University for a period from January 2009 to December 2020. The oral photographic images were captured from various oral cavity areas. The images were of varying resolutions, the largest was 4496 x 3000 pixels and the smallest was 1081 x 836 pixels. The dataset of 980 images was divided into 365 images of OSCC, 315 images of OPMDs and 300 images of non-

pathological oral images. The non-pathological oral images were defined as an image of oral mucosa which showed no pathological lesions, e.g., pigmented lesions, OPMDs and malignant lesions.

The reference data used in this study were clinical oral photographs of OSCC, OPMDs and non-pathological oral images which were located in various areas of the oral cavity including buccal mucosa, tongue, upper /lower alveolar ridge, floor of mouth, retromolar trigone and lip. All of the OSCC and OPMDs images were biopsy proven confirmed by oral pathologists as the gold standard for diagnosis. The OSCC images, which are OSCC stage I-IV according to the TNM clinical staging system as proposed by the American Joint Committee on Cancer (AJCC) [18], and OPMDs images used for analysis in this study were oral leukoplakia, erythroplakia, erythroleukoplakia, white striae and erythematous lesion surrounded with white striae with the pathological results of mild, moderate and severe epithelial dysplasia, hyperkeratosis and oral lichen planus.

## Experiment

All photographic images were uploaded to the VisionMarker server and web application for image annotation (Digital Storemesh, Bangkok, Thailand). The public version is available on GitHub (GitHub, Inc., CA, USA). The lesion boundaries of the OSCC and OPMDs images were annotated by three oral and maxillofacial surgeons. Owing to the differences in manual labeling from one surgeon to another, the ground truth used was the largest area of intersection between all of the surgeons' annotations in the CNN training, validation and testing (Fig 1).

## Image classification

Image classification refers to computer algorithms that can classify an image into a certain class according to its visual content. In this work, the CNN-based image classification networks, DenseNet-169, ResNet-101, SqueezeNet and Swin-S, were adopted to create the multiclass image classification model of "OSCC" and "OPMDs" apart from non-pathological oral images on oral photographic images. The image classification experiment was tested on Google Colab (Google Inc., CA, USA) using a Tesla P100, Nvidia driver: 460.32 and CUDA: 11.2 (Nvidia Corporation, CA, USA). The images were preprocessed by augmentation using Keras ImageDataGenerator (open-source software) then the framework resized input images to 224 x 224 pixels to feed into a neural network. The neural network architectures in this experiment are DenseNet-169, ResNet-101, SqueezeNet and Swin-S. DenseNet-169 and ResNet-101 are pre-trained weight from ImageNet except SqueezeNet and Swin-S which are pre-trained from scratch. The DenseNet-169, ResNet-101, SqueezeNet and Swin-S were modified to have 2-dimension output vectors, for multiclass: OSCC, OPMDs and non-pathological oral image, with softmax activation function. The hyper parameters used in this study were as follows: maximum number of epochs was 43, batch size of 32 and learning rate was 0.00001, except for Swin-S which had maximum number of epochs of 100 and batch size of 16. The validation loss was very close to the training loss, and there was no significant indication of over-fitting. The details of each image classification algorithm were as follows:

- Densely Connected Convolutional Networks (DenseNet) was proposed by Huang et al. [19] as a CNN-based classification algorithm which connects all layers (with matching feature-map sizes) directly with each other. DenseNet exploits the potential of the network through feature reuse, yielding condensed models that are easy to train and highly parameter efficient which is a good feature extractor for various computer vision tasks that build on convolutional features.

# Annotation

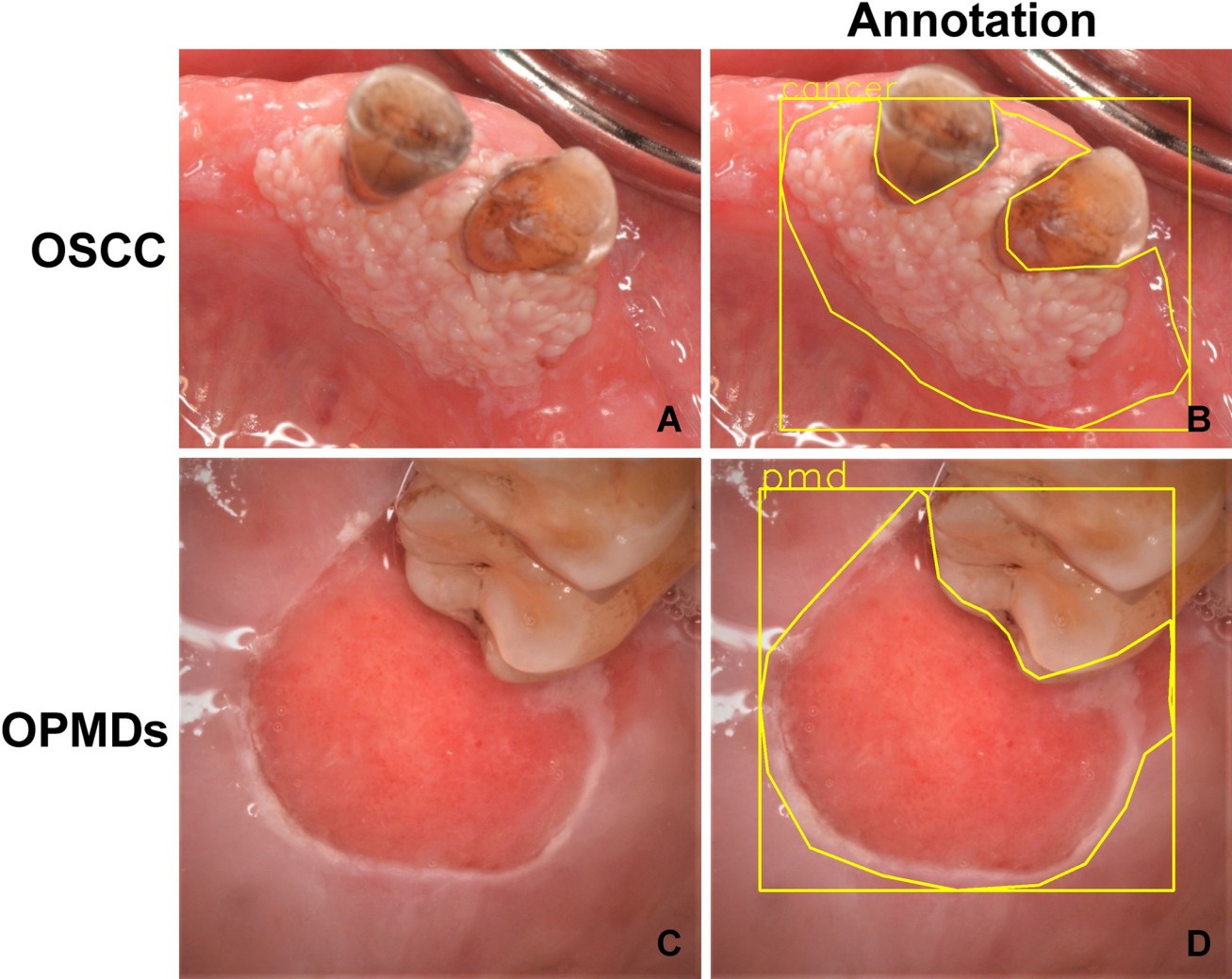

**Fig 1. Examples of the OSCC and OPMDs images from the dataset showing. (A)** OSCC image; **(B)** annotation of OSCC image by surgeons; **(C)** OPMDs image; **(D)** annotation of OPMDs image by surgeons.

- Residual Networks (ResNet) was developed by He et al. [20] as an architecture that is implemented by reformulating the layers as learning residual functions with reference to the layer inputs. This residual learning framework can gain more accuracy of object classification from considerably increased depth, producing results substantially better than previous networks.

- SqueezeNet was proposed by Iandola et al. [21] as a small CNN architecture with model compression techniques to less than 0.5 MB by decreasing the quantity of parameters and maximizing accuracy on a limited budget of parameters. SqueezeNet had 50x fewer parameters than a previous CNN, AlexNet, but maintained AlexNet-level accuracy.

- Swin Transformer (Swin) was presented by Liu et al. [22] as a new vision transformer which produces a hierarchical feature representation and has linear computational complexity with respect to input image size. The design of Swin as a shifted window based self-attention is shown to be effective and efficient on image classification.

## Object detection

Detection of lesions is another key to success in disease diagnosis. The CNN-based object detection is shown to be effective in identifying disease in the image. In this study, Faster R-CNN, YOLOv5, RetinaNet and CenterNet2 were adopted to detect the OSCC and OPMDs lesions in oral photographic images. The object detection experiment used the annotated image from VisionMarker (Digital Storemesh, Bangkok, Thailand). The annotated images were identified by bounding boxes showing locations of the lesion areas; then the pairs of image and annotation were ready for the training process. The image was preprocessed by augmentation using Keras ImageDataGenerator (open-source software) then the framework resized an input image to 256 x 256 pixels, except YOLOv5 which resized an input image to 640 x 640 pixels, to feed into a neural network. The training was performed on an on-premise server with 2 of GPU, TitanXP 12GB, Nvidia driver: 450.102 and CUDA: 11.0 (Nvidia Corporation, CA, USA). The neural network architectures were Faster-R-CNN, YOLOv5, RetinaNet and CenterNet2 with the pre-trained weight from COCO Detection. All the networks were trained using stochastic gradient descent (SGD). The hyper parameters used in this study were as follows: 20,000 iterations, maximum number of epochs was 1,882, learning rate of 0.0025 and batch size per image of 128, except for YOLOv5 which had maximum number of epochs of 200, learning rate of 0.01 and batch size per image of 8. The training loss was reduced and maintained between 15,000 and 20,000 iterations. The details of each object detection algorithm were as follows:

- Faster regional convolutional neural network (Faster R-CNN) was introduced by Ren et al. [23] as a CNN-based object detection framework. This algorithm is the combination of the previous object detection system, Fast R-CNN, and Region Proposal Networks (RPNs) into a single network to share their convolutional features leading to a more real-time object detection method. This design significantly improved the speed and accuracy in the object detection compared to basis R-CNN. Faster R-CNN is the very early object detection proposed to tackle both the localization and classification problems in a single deep learning network so the visual kernel can be computed once for both problems in a single deep neural network forward operation, also known as end-to-end. The input image has passed to CNN network such as VGG network to get the internal latent tensor (intermediate layer) then sends the tensor to two separate subnetworks; first subnetwork performing bounding box location regression and also computing the classification in the second subnetwork. Where the loss function is defined as

- $L = \frac{1}{N_{cls}} \sum_i Lcls_i + \lambda \frac{1}{N_{reg}} \sum_i Lreg_i$, where L is the total loss, i is the index of an anchor in a mini-batch, Ncls is the number of possible sub-image from sliding window, Lcls is log loss of classification, $\lambda$ is a hyperparameter to balance the two loss functions, Nreg is the number of anchor locations and Lreg is a loss function for location regression computed from the robust loss function (smooth L1) [24].

- You only look once (YOLO) was proposed by Redmon et al. [25] as a CNN-based object detection algorithm which reframes as a single regression problem, straight from image pixels to bounding box coordinates and class probabilities. The YOLO design enables end-to-end training and realtime speeds while maintaining high average precision. Due to early success of Faster R-CNN in terms of high accuracy baseline, YOLO tackled another aspect of object deletion problem by dramatically increasing the frame-rate at 45 frames per second on a Titan X GPU (Nvidia Corporation, CA, USA). The intersection over union metric (IoU) is emphasized in this work to make the region proposal generation bounding box location more accurate by reframing object detection as a single regression problem,

straight from image pixels to bounding box coordinates and class probabilities resulting in less computation and having high frame rate performance.

- RetinaNet was proposed by Lin et al. [26] as a simple one-stage object detector with a new loss function that acts as a more effective alternative to previous algorithms for dealing with class imbalance. This design achieves state of-the-art accuracy and speed for the object detection. Introduced a novel loss function by adding Focal Loss function to original cross entropy to improve accuracy of dense object detectors. Furthermore the RetinaNet architecture adopts Feature Pyramid Network (FPN), which is based on top-down pathway to allow the top level feature to laterally connect to the feature extraction of each layer leading to multi scale feature extraction capability therefore the RetinaNet able to detect smallest and biggest objects effectively.

- CenterNet2 was developed by Zhou et al. [27] as a probabilistic interpretation of two-stage detectors. This algorithm was designed as a simple modification of standard two-stage detector training by optimizing a lower bound to a joint probabilistic objective over both stages which achieved desirable speed and accuracy for the object detection. The CentetNet revisited the two stage object detection model, where the first stage is to compute the probability of an object in the observation image also called object likelihood to get the bounding box and the second step is to classify the object. The major difference of the CenterNet2 is applying object likelihood and conditional probability to classification $P(C_k) = P(C_k|O_k)P(O_k)$, where k is index of detection bounding box $P(O_k)$ is first-stage object likelihood, $P(C_k|O_k)$ is conditional probability the given object be the class $C_k$ and $P(C_k)$ is the probability of bounding box k be the class $C_k$.

To evaluate the performance of the image classification and object detection networks, five-fold cross-validation was employed. Data elements were split into 5 subsets using random sampling with equal numbers of OSCC, OPMDs and non-pathological oral images. Then, one subset was considered as a testing set, while the remaining four subsets were used as training and validation sets. This process was repeated 5 times to involve all subsets as testing sets.

## Evaluation measures

The metrics used to evaluate the machine learning algorithms in bioinformatics were used in this study [28]. The CNN-based image classification models were evaluated using the precision, recall (sensitivity), specificity, F1 score, and area under the receiver operating characteristics curve (AUC of ROC) to measure the performance in classifying OSCC and OPMDs on the oral photographic images. The classification performance of models was also evaluated by generating a heat map visualization using the gradient-weighted class activation mapping (Grad-CAM) [29] to see how the models classify and identify OSCC and OPMDs on photographic images. For the object detection, the performance of the CNN-based object detection models was evaluated to detect a bounding box relative to the ground truth region in the OSCC and OPMDs images by the precision, recall, F1 score and AUC of precision-recall curve. If the IoU value between the generated bounding box and the ground truth was less than 0.5, then the produced bounding box was considered to be a false prediction or false positive.

A test dataset with known pathological results was evaluated to compare the performance of the CNN-based classification models with that of 20 clinicians; 10 board certified oral and maxillofacial surgeons and 10 GPs who have at least 2 years of experience in dental practice in rural hospitals. None of these readers participated in the clinical care or assessment of the enrolled patients, nor did they have access to their medical records. The overall sensitivity and specificity of these clinicians were calculated. Data analyses were conducted using SPSS

version 22.0 (SPSS, Chicago, IL). The statistical analysis for image classification and object detection was calculated as follows:

$$IoU = \text{area of overlap}/\text{area of union}$$

$$Precision = TP/TP + FP$$

$$Recall\,(Sensitivity) = TP/TP + FN$$

$$Specificity = TN/TN + FP$$

$$F1\,score = 2\,x\,(Precision\,x\,Recall)/Precision + Recall$$

- True positive (TP): positive outcomes that the model predicted correctly which $IoU > 0.5$.
- False positive (FP): positive outcomes that the model predicted incorrectly which $IoU < 0.5$.
- True negative (TN): negative outcomes that the model predicted correctly.
- False negative (FN): negative outcomes that the model predicted incorrectly.

## Results

### Image classification results

The evaluation of multiclass images was performed on the test set and the results of the CNN-based image classification models are reported in Table 1. The image classification of CNN-based image classification models achieved a precision, a recall (sensitivity), a specificity, an F1 score and AUC of ROC curve as seen in Table 2. The overall sensitivity and specificity for the classification by the ten oral and maxillofacial surgeons of OCSS were 0.90 (95%CI = 0.85–0.96) and 0.89 (95%CI = 0.81–0.97) and OPMDs were 0.74 (95%CI = 0.61–0.87) and 0.93 (95%CI = 0.90–0.96), respectively. In addition, the overall sensitivity and specificity for the classification by the ten GPs of OCSS were 0.77 (95%CI = 0.70–0.85) and 0.87 (95%CI = 0.85–0.90) and OPMDs were 0.68 (95%CI = 0.62–0.75) and 0.86 (95%CI = 0.82–0.90), respectively.

**Table 1. Multi-class image classification performances of CNN algorithms on the test sets compared with the average performance of clinicians; 'oral and maxillofacial surgeons' vs. 'GPs'.**

|  | Class | | | | | | | | | |
|---|---|---|---|---|---|---|---|---|---|---|
|  | OSCC | | | | | OPMDs | | | | |
|  | Precision | Recall (Sensitivity) | Specificity | F1 score | AUC of ROC curve | Precision | Recall (Sensitivity) | Specificity | F1 score | AUC of ROC curve |
| DenseNet-169 | 0.98 | 0.99 | 0.99 | 0.98 | 1.0 | 0.95 | 0.95 | 0.97 | 0.95 | 0.98 |
| ResNet-101 | 0.96 | 0.92 | 0.94 | 0.94 | 0.99 | 0.97 | 0.97 | 0.94 | 0.97 | 0.97 |
| SqueezeNet | 0.85 | 0.72 | 0.92 | 0.78 | 0.88 | 0.76 | 0.78 | 0.88 | 0.77 | 0.87 |
| Swin-S | 0.69 | 0.73 | 0.83 | 0.71 | 0.71 | 0.63 | 0.74 | 0.88 | 0.68 | 0.80 |
| Oral and maxillofacial surgeons | - | 0.90 | 0.89 | - | - | - | 0.74 | 0.93 | - | - |
| GPs | - | 0.77 | 0.87 | - | - | - | 0.68 | 0.86 | - | - |

AUC, area under the curve; ROC, receiver operating characteristics; GPs, General practitioners.

**Table 2. Multi-class object detection performances of CNN algorithms on the test sets.**

| | Class | | | | | | | |
|---|---|---|---|---|---|---|---|---|
| | OSCC | | | | OPMDs | | | |
| | Precision | Recall (Sensitivity) | F1 score | AUC of precision—recall curve | Precision | Recall (Sensitivity) | F1 score | AUC of precision—recall curve |
| Faster R-CNN | 0.84 | 0.90 | 0.87 | 0.88 | 0.60 | 0.71 | 0.65 | 0.64 |
| YOLOv5 | 0.88 | 0.86 | 0.87 | 0.84 | 0.74 | 0.39 | 0.51 | 0.34 |
| RetinaNet | 0.98 | 0.82 | 0.89 | 0.81 | 0.92 | 0.57 | 0.70 | 0.55 |
| CenterNet2 | 0.64 | 0.92 | 0.76 | 0.91 | 0.49 | 0.60 | 0.54 | 0.58 |
| Oral and maxillofacial surgeons | - | 0.90 | - | - | - | 0.74 | - | - |
| GPs | - | 0.77 | - | - | - | 0.68 | - | - |

AUC, area under the curve; GPs, General practitioners

Fig 2 shows an example of the Grad-CAM visualization of the DenseNet-169 output of OSCC and OPMDs classes which shows that the model correctly classifies and identifies OSCC and OPMDs on photographic images.

## Object detection results

The object detection models were evaluated on the test set and the results are reported in Table 2. The detection performance of CNN-based object detection models achieved a precision, a recall, an F1 score and AUC of precision-recall curve as shown in Table 2. Examples of detection outputs from CNN-based object detection models in this study are provided in Fig 3.

## Discussion

Oral cancer screening is an important part of an oral examination, the goal of which is to identify changes and the development of oral cancer. It is commonly known that OSCC, the most common oral cancer, is often preceded by OPMDs [11]. Patients with oral lesions are often first seen by GPs, both medical and dental. Therefore, GPs are in a unique position to detect oral cancer at early stages. Nevertheless, several studies indicated that the GPs' s lack knowledge and awareness in the area of oral cancer diagnosis, especially an early sign of oral cancer, is the most significant factor in delaying referral and treatment of oral cancer [30, 31]. Delay in diagnosing oral cancer may lead to more invasive treatment resulting in greater morbidity of oral functions, such as distortions of speech, chewing and swallowing, which will have a significant impact on individual's quality of life [13]. Usually, when diagnosed at an advanced stage, less than 50% of oral cancer patients survive more than 5 years. This rate has remained disappointingly low and relatively constant during the last few decades [10, 15]. Therefore, the early detection of oral cancer, especially OPMDs or early stage OSCC, with appropriate referral to specialists is crucial to control the disease and improve the survival rate and quality of life of patients. Screening of oral cancer is largely based on visual examination. The current adjunctive diagnostic aids for oral cancer screening include oral cytology, vital staining with toluidine blue and light detector systems, e.g., VELscope. But no technology provides definitive evidence to suggest that it improves the sensitivity or specificity of oral cancer screening beyond oral examination [32]. In recent year, AI techniques have improved performance in areas of image analysis with a range of promising applications in medicine. The flood of medical data in the form of image data and learning algorithms is accelerating the development of AI-based image

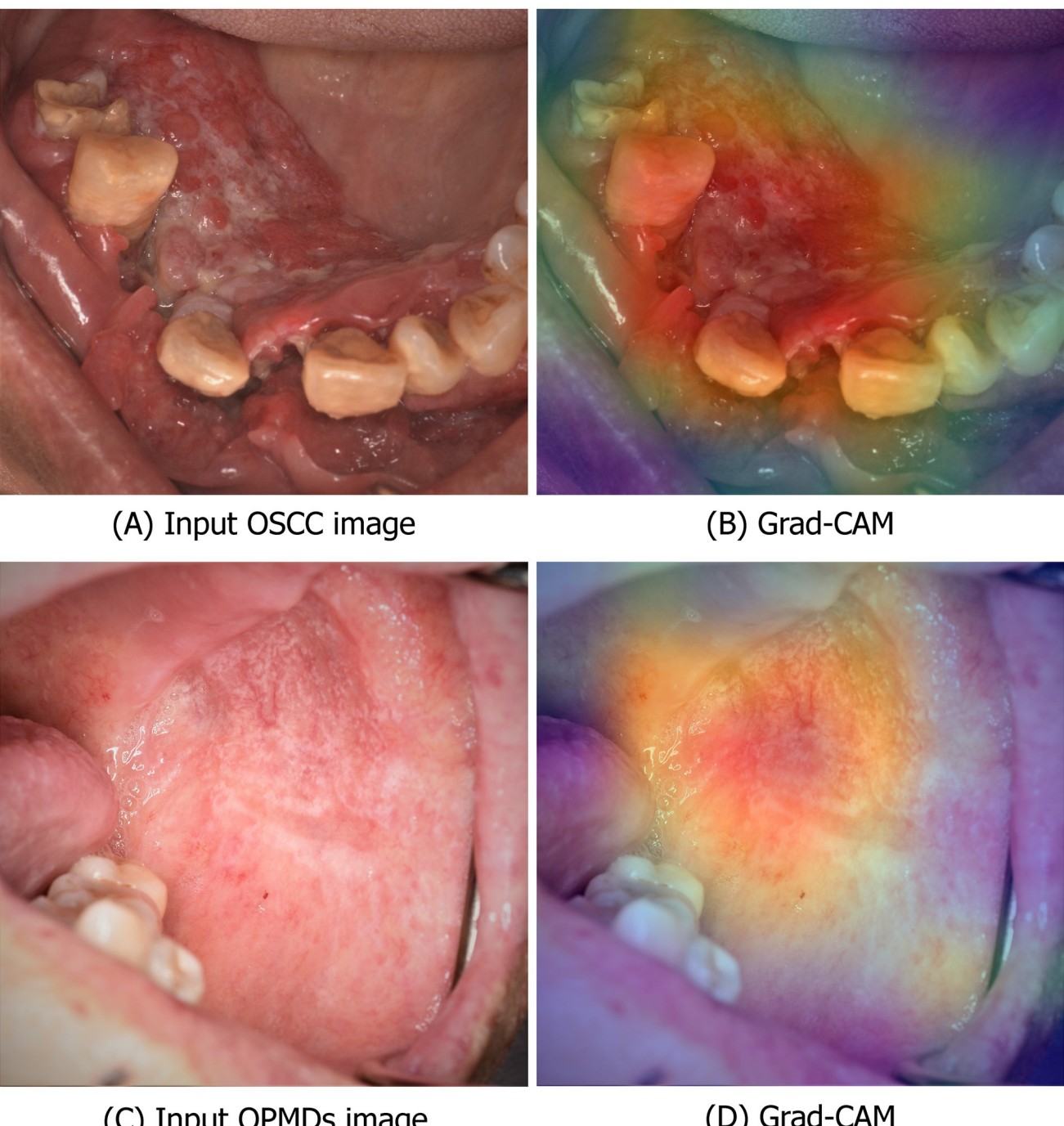

**(A) Input OSCC image**

**(B) Grad-CAM**

**(C) Input OPMDs image**

**(D) Grad-CAM**

**Fig 2. Example of the Grad-CAM visualization of the DenseNet-169. (A)** Image with OSCC lesion; **(B)** The model correctly classified OSCC and labeled the correct location. **(C)** Image with OPMDs lesion **(D)** The model correctly classified OPMDs and labeled the correct location.

analysis that promises to improve efficiency, effectiveness and speed of diagnosis enabling new insights about diagnoses, treatment options and patient outcomes [33]. Advances in computer vision and AI technology that improve visual detection can be used to assist visual examination combined with clinical data as a novel diagnostic tool in the oral cancer screening system.

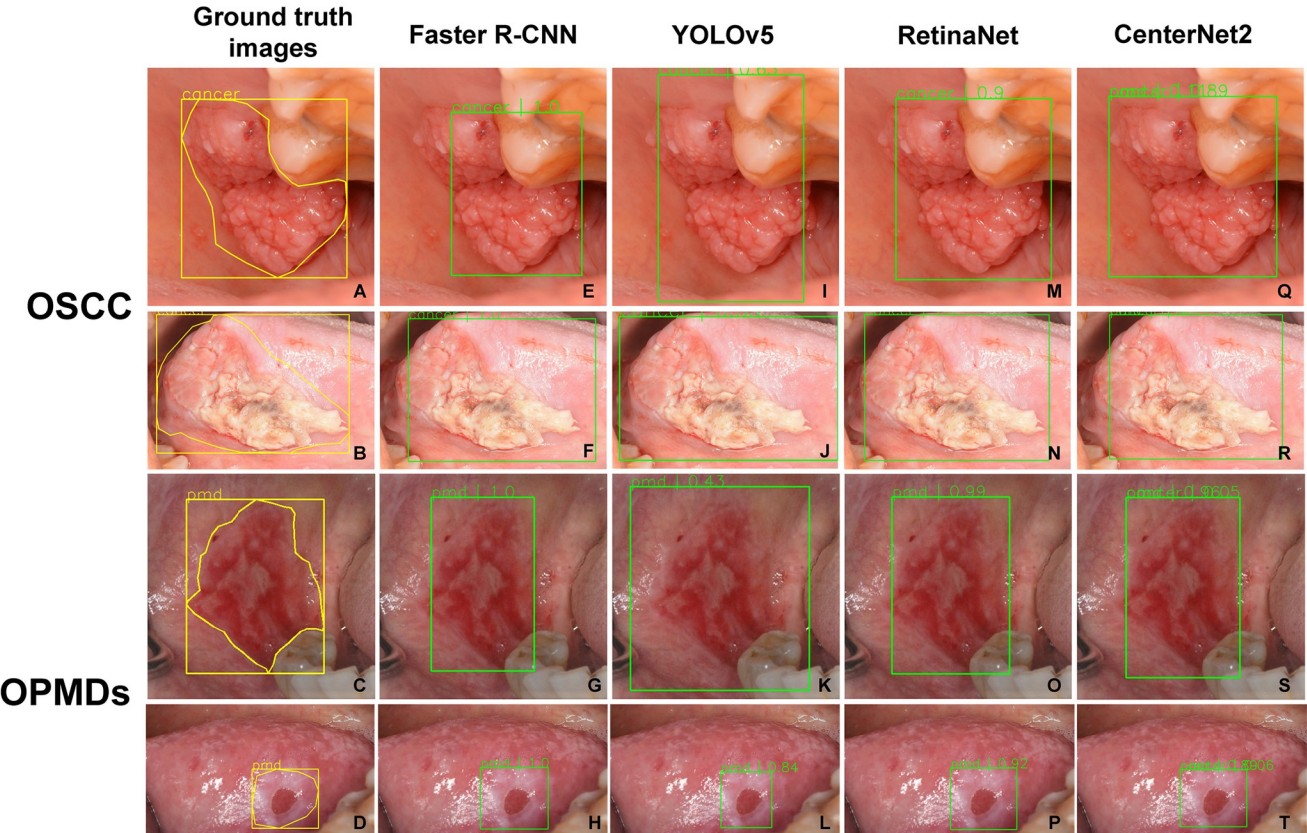

**Fig 3. (A-B)** Bounding box ground truth based on surgeons' annotations of the imaging of the patient with OSCC at retromolar trigone and lateral tongue, respectively; **(C-D)** Bounding box ground truth based on surgeons' annotations of the imaging of the patient with OPMDs at retromolar trigone and lateral tongue, respectively; **(E-H)** The true positive outputs from the faster R-CNN detection; **(I-L)** The true positive outputs from the YOLOv5 detection; **(M-P)** The true positive outputs from the RetinaNet detection; **(Q-T)** The true positive outputs from the CenterNet2 detection.

In this work, the performance of CNN-based image classification models works well to identify the OSCC and OPMDs. The results, particularly in DenseNet-169 and ResNet-101, achieved near-perfect AUC and showed performance similar to the classification of multiclass image of OSCC and OPMDs on oral images as a CNN model of the study of Fu et al. [34], Tan-river et al. [35] and Song et al. [36] but more accurate than the studies of Welikala et al. [37]. The difference in the performance of models may be from variations in the class distribution of each study. DenseNet-169 and ResNet-101 are a series of well-optimized algorithms, which achieve high performance in image classification, and are widely used in the medical field. However, the DenseNet-169 and ResNet-101 algorithms were a large CNN architecture and required a high-performance computing server for the image classification processing which may not be appropriate for use in a mobile application for oral cancer screening. Therefore, this work selected new and smaller CNN models, SqueezeNet and Swin-S, to test the classification performance of OSCC and OPMDs on oral photographic images. SqueezeNet and Swin-S showed acceptable accuracy and achieved an AUC of 0.71–0.88 which may have inferior performance than DenseNet-169 and ResNet-101. But the small size architecture of these models was more suited for developing into a mobile application for oral cancer screening. In the medical field, there was a study that successfully used SqueezeNet for the diagnosis of the coronavirus disease 2019 (COVID-19) from chest X-ray images [38]. To the best of our knowledge, this is the first study to use Swin-S for classification of oral lesions. Previous studies [35–37, 39,

40] have demonstrated the potential for classification performance of various CNN-based algorithms without comparison with the clinician's clinical diagnostic decision of oral lesions on photographic images. The strength of this study was the use of histopathologic determination as the ground truth. The results showed that these CNN-based classification models yield a classification performance of OSCC and OPMDs on oral photograph equal to expert level (board certified oral and maxillofacial surgeons) and superior to GP level. Moreover, DenseNet-169 and ResNet-101 even outperformed expert-level classification performance.

For the detection of oral lesions, the CNN-based object detection used in this study showed good performance in the detection of OSCC and OPMDs on photographic images which achieved AUC of 0.81–0.91 and 0.34–0.64 in the detection of OSCC and OPMDs, respectively. One of the generally CNN-based object detection algorithms used in medical images, the faster R-CNN achieved high performance in the detection of OSCC and OPMDs with AUC of 0.88 and 0.64, respectively. The faster R-CNN detection performance in this study achieved higher precision, recall and F1 score than the previous study of Welikala et al. [37] for detecting the OSCC and OPMDs on oral photographs which may be from the different number of classes in the study. Nowadays, there is a continuous development of CNN-based object detection to increase the accuracy of detection of the interested object. CenterNet2, one of the latest CNN-based object detection models, achieved the highest performance in detection of OSCC, an AUC of 0.91, but was slightly inferior to faster R-CNN for detecting of OPMDs. The overall OPMDs performance in detection in this study was not as good as the detection of OSCC which may result from the general characteristics of OPMDs in the oral cavity which make them difficult to recognize, even by the expert. The lowest performance model in detection of OPMDs is YOLOv5 which achieved a precision of 0.34, a recall of 0.39, a F1 score of 0.51 and an AUC of 0.34. Even so the results were comparable to those of the study by Tanriver et al. [35] This may be due to YOLOv5 being an extremely fast detection model with an operating time of only 0.07 seconds per frame [25]. A high-speed model of this type may not be appropriate for detecting the features of OPMDs on oral images.

Deep CNN models have potential for binary classification and detection of OPMDs [39] and OSCC [40] in oral photographs. In the real-world scenario, the clinical characteristics of OPMDs can show considerable variation which can mimic the likelihood of malignancy, and *vice versa*. In this regard, multiclass classification and object detection were explored using several CNN-based algorithms in this study. The AUC of the best multiclass CNN models yielded results comparable to those of binary classification and detection.

As the focus of AI is shifting from model/algorithm development to the quality of the data used to train the models [41], this study has limitations that need to be addressed. First, the dataset was small and only included OSCC and OPMDs images. And second, the process of labeling lesions on oral photographic images required experts to identify the ground truth on the images, which was time consuming. For future work, we plan to develop the CNN-based mobile application to collect more data and expand the image dataset to include other oral lesions such as pigmented lesions and submucosal lesions, from the multi-cancer center and hospitals in a remote area. In addition, we plan to develop the system integrated into the clinical workflow to allow the experts to label the ground truth of the lesion in the image. This not only saves time on the labeling process, but also increases the chances of the experts to thoroughly study the details of the lesion in the image.

## Conclusions

CNN-based models showed comparable diagnostic performances to expert level in classifying OSCC and OPMDs on oral photographic images. In particular, DenseNet-169 and ResNet-

101 even outperformed expert-level classification performance. This is expected to be a novel innovation as a diagnostic tool to assist clinicians, especially GPs, in improving the accuracy of early detection of cancerous lesions and support expert-level decision making in the oral cancer screening program.

## Supporting information

**S1 Fig. The receiver operating characteristic (ROC) curve of high performance CNN-based multiclass classification models.**
(PDF)

**S2 Fig. Normalized confusion matrix of high performance CNN-based multiclass classification models.**
(PDF)

**S3 Fig. The precision-recall curve of CNN-based object detection models.**
(PDF)

**S1 File. Object detection matrix of CNN-based object detection models.**
(PDF)

## Acknowledgments

We gratefully acknowledge the support of the NVIDIA Corporation for the Titan Xp GPU (Nvidia Corporation, CA, USA) used in this research. We thank Waranthorn Chansawang for their assistance with the deep learning model training.

## Author Contributions

**Conceptualization:** Kritsasith Warin, Wasit Limprasert, Siriwan Suebnukarn.

**Data curation:** Kritsasith Warin, Wasit Limprasert, Patcharapon Jantana, Sothana Vicharueang.

**Formal analysis:** Kritsasith Warin, Wasit Limprasert, Patcharapon Jantana, Sothana Vicharueang.

**Funding acquisition:** Kritsasith Warin.

**Investigation:** Wasit Limprasert, Patcharapon Jantana, Sothana Vicharueang.

**Methodology:** Kritsasith Warin, Wasit Limprasert, Siriwan Suebnukarn.

**Project administration:** Kritsasith Warin.

**Resources:** Kritsasith Warin, Suthin Jinaporntham.

**Software:** Wasit Limprasert.

**Supervision:** Siriwan Suebnukarn.

**Writing – original draft:** Kritsasith Warin, Siriwan Suebnukarn.

**Writing – review & editing:** Kritsasith Warin, Siriwan Suebnukarn.

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
