## [Decision Letter · Decision Letter 0]

27 Apr 2022

PONE-D-22-06226AI-based analysis of oral lesions using novel deep convolutional neural networks for early detection of oral cancerPLOS ONE

Dear Dr. Warin,

Thank you for submitting your manuscript to PLOS ONE. After careful consideration, we feel that it has merit but does not fully meet PLOS ONE’s publication criteria as it currently stands. Therefore, we invite you to submit a revised version of the manuscript that addresses the points raised during the review process.

We look forward to receiving your revised manuscript.

Kind regards,

Ayan Seal, Ph.D

Academic Editor

PLOS ONE

Journal Requirements:

Additional Editor Comments (if provided):

Reviewers' comments:

Reviewer's Responses to Questions

**Comments to the Author**

1. Is the manuscript technically sound, and do the data support the conclusions?

Reviewer #1: Yes

Reviewer #2: Yes

2. Has the statistical analysis been performed appropriately and rigorously? 

Reviewer #1: Yes

Reviewer #2: Yes

3. Have the authors made all data underlying the findings in their manuscript fully available?

Reviewer #1: Yes

Reviewer #2: Yes

4. Is the manuscript presented in an intelligible fashion and written in standard English?

Reviewer #1: Yes

Reviewer #2: Yes

5. Review Comments to the Author

Reviewer #1: 1.The study presents the results of primary scientific research.

2.Results reported have not been published elsewhere.

3.Experiments, statistics, and other analyses are performed to a high technical standard and are described in sufficient detail.

4.Conclusions are presented in an appropriate fashion and are supported by the data.

5.The article is presented in an intelligible fashion and is written in standard English.

6.The research meets all applicable standards for the ethics of experimentation and research integrity.

7.The article adheres to appropriate reporting guidelines and community standards for data availability.

Reviewer #2: Briefly try to describe the models which are used in object detection by what all layers they comprise of, and in what fashion.

If possible try to highlight the importance of one or more layers and in what way they are supportive for your case.

Also try to give some mathematical expression/equations support to your research.

6. PLOS authors have the option to publish the peer review history of their article (what does this mean?). If published, this will include your full peer review and any attached files.

Reviewer #1: No

Reviewer #2: **Yes: **Dr. Rajashekhargouda C. Patil

---

## [Author Response · Author response to Decision Letter 0]

5 Jul 2022

Respond to Reviewers

Journal Requirements:

Response: Thanks for your suggestion, we checked the references to ensure they meet the requirements of the journal.

Additional Editor Comments (if provided):

Reviewers' comments:

Reviewer's Responses to Questions

Comments to the Author

1. Is the manuscript technically sound, and do the data support the conclusions?

Reviewer #1: Yes

Reviewer #2: Yes

2. Has the statistical analysis been performed appropriately and rigorously?

Reviewer #1: Yes

Reviewer #2: Yes

3. Have the authors made all data underlying the findings in their manuscript fully available?

Reviewer #1: Yes

Reviewer #2: Yes

4. Is the manuscript presented in an intelligible fashion and written in standard English?

Reviewer #1: Yes

Reviewer #2: Yes

5. Review Comments to the Author

Reviewer #1: 1.The study presents the results of primary scientific research.

2.Results reported have not been published elsewhere.

3.Experiments, statistics, and other analyses are performed to a high technical standard and are described in sufficient detail.

4.Conclusions are presented in an appropriate fashion and are supported by the data.

5.The article is presented in an intelligible fashion and is written in standard English.

6.The research meets all applicable standards for the ethics of experimentation and research integrity.

7.The article adheres to appropriate reporting guidelines and community standards for data availability.

Response: We appreciated the reviewer for the comments.

Reviewer #2: Briefly try to describe the models which are used in object detection by what all layers they comprise of, and in what fashion.

If possible try to highlight the importance of one or more layers and in what way they are supportive for your case.

Also try to give some mathematical expression/equations support to your research.

Response: We thank the reviewer for the comment, we have added more detail on the object detection algorithms layer, more reference (ref 24) and mathematical expressions/equations in the Object detection subsection of the Materials and Methods section.

Line 181: Faster R-CNN is the very early object detection proposed to tackle both the localization and classification problems in a single deep learning network so the visual kernel can be computed once for both problems in a single deep neural network forward operation, also known as end-to-end. The input image has passed to CNN network such as VGG network to get the internal latent tensor (intermediate layer) then sends the tensor to two separate subnetworks; first subnetwork performing bounding box location regression and also computing the classification in the second subnetwork. Where the loss function is defined as 

L=(1)/Ncls ∑_i▒〖Lcls〗_i + λ(1)/Nreg ∑_i▒〖Lreg〗_i , where L is the total loss, i is the index of an anchor in a mini-batch, Ncls is the number of possible sub-image from sliding window, Lcls is log loss of classification, λ is a hyperparameter to balance the two loss functions, Nreg is the number of anchor locations and Lreg is a loss function for location regression computed from the robust loss function (smooth L1) [24].

 Reference 24: Girshick R, editor Fast R-CNN. 2015 IEEE International Conference on Computer Vision (ICCV); 2015 7-13 Dec. 2015.

Line 196: Due to early success of Faster R-CNN in terms of high accuracy baseline, YOLO tackled another aspect of object deletion problem by dramatically increasing the frame-rate at 45 frames per second on a Titan X GPU (Nvidia Corporation, CA, USA). The intersection over union metric (IoU) is emphasized in this work to make the region proposal generation bounding box location more accurate by reframing object detection as a single regression problem, straight from image pixels to bounding box coordinates and class probabilities resulting in less computation and having high frame rate performance.

Line 206: Introduced a novel loss function by adding Focal Loss function to original cross entropy to improve accuracy of dense object detectors. Furthermore the RetinaNet architecture adopts Feature Pyramid Network (FPN), which is based on top-down pathway to allow the top level feature to laterally connect to the feature extraction of each layer leading to multi scale feature extraction capability therefore the RetinaNet able to detect smallest and biggest objects effectively.

Line 215: The CentetNet revisited the two stage object detection model, where the first stage is to compute the probability of an object in the observation image also called object likelihood to get the bounding box and the second step is to classify the object. The major difference of the CenterNet2 is applying object likelihood and conditional probability to classification P(〖C〗_k)=P(〖C〗_k|〖O〗_k)P(〖O〗_k), where k is index of detection bounding box P(〖O〗_k) is first-stage object likelihood, P(〖C〗_k|〖O〗_k) is conditional probability the given object be the class 〖C〗_k and P(〖C〗_k) is the probability of bounding box k be the class 〖C〗_k.

---

## [Decision Letter · Decision Letter 1]

10 Aug 2022

AI-based analysis of oral lesions using novel deep convolutional neural networks for early detection of oral cancer

PONE-D-22-06226R1

Dear Dr. Warin,

We’re pleased to inform you that your manuscript has been judged scientifically suitable for publication and will be formally accepted for publication once it meets all outstanding technical requirements.

Kind regards,

Ayan Seal, Ph.D

Academic Editor

PLOS ONE

Additional Editor Comments (optional):

Reviewers' comments:

Reviewer's Responses to Questions

**Comments to the Author**

1. If the authors have adequately addressed your comments raised in a previous round of review and you feel that this manuscript is now acceptable for publication, you may indicate that here to bypass the “Comments to the Author” section, enter your conflict of interest statement in the “Confidential to Editor” section, and submit your "Accept" recommendation.

Reviewer #2: All comments have been addressed

2. Is the manuscript technically sound, and do the data support the conclusions?

Reviewer #2: Yes

3. Has the statistical analysis been performed appropriately and rigorously? 

Reviewer #2: Yes

4. Have the authors made all data underlying the findings in their manuscript fully available?

Reviewer #2: Yes

5. Is the manuscript presented in an intelligible fashion and written in standard English?

Reviewer #2: Yes

6. Review Comments to the Author

Reviewer #2: (No Response)

7. PLOS authors have the option to publish the peer review history of their article (what does this mean?). If published, this will include your full peer review and any attached files.

Reviewer #2: **Yes: **Rajashekhargouda C. Patil

---

## [Editor Report · Acceptance letter]

16 Aug 2022

PONE-D-22-06226R1 

AI-based analysis of oral lesions using novel deep convolutional neural networks for early detection of oral cancer 

Dear Dr. Warin:

I'm pleased to inform you that your manuscript has been deemed suitable for publication in PLOS ONE. Congratulations! Your manuscript is now with our production department. 

Kind regards, 

on behalf of

Dr. Ayan Seal 

Academic Editor

PLOS ONE